# Laurequinone, a Lead Compound against *Leishmania*

**DOI:** 10.3390/md21060333

**Published:** 2023-05-30

**Authors:** Sara García-Davis, Atteneri López-Arencibia, Carlos J. Bethencourt-Estrella, Desirée San Nicolás-Hernández, Ezequiel Viveros-Valdez, Ana R. Díaz-Marrero, José J. Fernández, Jacob Lorenzo-Morales, José E. Piñero

**Affiliations:** 1Instituto Universitario de Bio-Orgánica Antonio González (IUBO AG), Universidad de La Laguna (ULL), Avenida Astrofísico Francisco Sánchez 2, 38206 La Laguna, Tenerife, Spain; sgdavis@ull.edu.es (S.G.-D.); adiazmar@ipna.csic.es (A.R.D.-M.); jjfercas@ull.edu.es (J.J.F.); 2Departamento de Química Orgánica, Universidad de La Laguna, Avenida Astrofísico Francisco Sánchez 2, 38206 La Laguna, Tenerife, Spain; 3Instituto Universitario de Enfermedades Tropicales y Salud Pública de Canarias, Universidad de La Laguna, Avenida Astrofísico Francisco Sánchez S/N, 38206 La Laguna, Tenerife, Spain; cbethene@ull.edu.es (C.J.B.-E.); dsannico@ull.edu.es (D.S.N.-H.); jmlorenz@ull.edu.es (J.L.-M.); jpinero@ull.edu.es (J.E.P.); 4Consorcio Centro de Investigación Biomédica en Red de Enfermedades Infecciosas (CIBERINFEC), |Instituto de Salud Carlos III, 28006 Madrid, Madrid, Spain; 5Departamento de Obstetricia y Ginecología, Pediatría, Medicina Preventiva y Salud Pública, Toxicología, Medicina Legal y Forense y Parasitología, Universidad de La Laguna, 38200 La Laguna, Tenerife, Spain; 6Facultad de Ciencias Biológicas, Universidad Autónoma de Nuevo León, Avenida Pedro de Alba S/N, San Nicolás de los Garza 66450, Nuevo León, Mexico; jose.viverosvld@uanl.edu.mx; 7Instituto de Productos Naturales y Agrobiología (IPNA), Consejo Superior de Investigaciones Científicas (CSIC), Avenida Astrofísico Francisco Sánchez 3, 38206 La Laguna, Tenerife, Spain

**Keywords:** laurequinone, sesquiterpene, *Laurencia*, *Leishmania amazonensis*, leishmaniasis

## Abstract

Among neglected tropical diseases, leishmaniasis is one of the leading causes, not only of deaths but also of disability-adjusted life years. This disease, caused by protozoan parasites of the genus *Leishmania,* triggers different clinical manifestations, with cutaneous, mucocutaneous, and visceral forms. As existing treatments for this parasitosis are not sufficiently effective or safe for the patient, in this work, different sesquiterpenes isolated from the red alga *Laurencia johnstonii* have been studied for this purpose. The different compounds were tested in vitro against the promastigote and amastigote forms of *Leishmania amazonensis*. Different assays were also performed, including the measurement of mitochondrial potential, determination of ROS accumulation, and chromatin condensation, among others, focused on the detection of the cell death process known in this type of organism as apoptosis-like. Five compounds were identified that displayed leishmanicidal activity: laurequinone, laurinterol, debromolaurinterol, isolaurinterol, and aplysin, showing IC_50_ values against promastigotes of 1.87, 34.45, 12.48, 10.09, and 54.13 µM, respectively. Laurequinone was the most potent compound tested and was shown to be more effective than the reference drug miltefosine against promastigotes. Different death mechanism studies carried out showed that laurequinone appears to induce programmed cell death or apoptosis in the parasite studied. The obtained results underline the potential of this sesquiterpene as a novel anti-kinetoplastid therapeutic agent.

## 1. Introduction

Neglected tropical diseases (NTDs) are a diverse group, comprising 20 diseases that affect more than 1 billion people who live mostly in tropical and subtropical impoverished communities. These diseases cause devastating health, social, and economic consequences, representing a public health challenge [1]. Among NTDs, leishmaniasis is one of the top 10 diseases, with more than 12 million infected people, 0.9 to 1.6 million new cases each year, and between 20,000 and 30,000 deaths per year [2]. Leishmaniasis is a vector-borne disease with a wide variety of parasite species and reservoirs involved. The disease is transmitted through the bite of infected female sandflies and manifests as cutaneous (CL), mucosal, and visceral (VL) forms. The CL form is the most common, whereas VL is the most severe [2].

Current chemotherapeutic options such as amphotericin B, miltefosine, paromomycin, pentamidine, and azole antifungals are available, and in recent years, advances in the understanding of the pathogenesis have resulted in the development of newer treatments such as immunotherapy and immunochemotherapy. However, their adverse effects, costs, treatment duration, and the emergence of drug resistance demonstrate the necessity of new treatment options [3]. A good example of this is miltefosine, currently the only oral drug available for the treatment of leishmaniasis. As for its antiparasitic activity, its mode of action has not yet been elucidated, but several hypotheses have emerged, such as induction of apoptosis, alteration of lipid-dependent cell signaling pathways, alteration of membrane composition, and immunomodulatory effects. In addition, miltefosine can depolarize the mitochondrial membrane potential and inhibit cytochrome c oxidase, which is related to parasite death by apoptosis [4,5]. Nevertheless, it also produces serious side effects, such as nephrotoxicity, hepatotoxicity, and severe gastrointestinal teratogenicity [6].

Historically, natural products have been used in the treatment of tropical parasitic diseases, either in their endogenous form or as lead compounds acting as inspiration for the development of therapeutic agents. From 1981 to 2019, 20 antiparasitic chemical entities were described from natural products and/or synthetic variations using their structures [7]. Especially, the marine environment is a well-established source of bioactive natural products, which have been reported for their antileishmanial, antimalarial, anti-plasmodial, anti-trypanosoma, and anti-amoebic properties. Besides the increasingly marketed marine-derived drugs, with 17 approved drugs against cancer, pain, hypertriglyceridemia, and viral infections, there are no antiparasitic agents, nor in clinical phases [8].

Natural product skeletons have re-emerged in drug discovery research projects, offering a different resource for finding new chemical entities with new modes of action. The marine environment offers a great chemical diversity of natural products, and thus great potential for drug discovery. Many of these compounds, which probably evolved as chemical defenses of the organism producing them, could serve as inspiration for the development of new antiprotozoal chemical entities. While this goal is clearly within reach, few natural compounds of marine origin have been tested in animal models of infection. Despite these challenges, the appeal of these nature-optimized leads has not diminished, and with improved access to both natural compounds and semi-synthetic derivatives, the likelihood of success in developing nature-inspired therapies is increasing [9].

Among marine sources, the study of algal extracts has resulted in the description of antiparasitic active compounds comprising chemical classes such as terpenes, sulphated polysaccharides, acetogenins, polyphenols, and others [10]. Accordingly, we have focused our research on the study of the genus *Laurencia*, a red alga that has been considered one of the most prolific sources of chemo-diversity among seaweed. Looking for new antiparasitic agents, we have identified active compounds from *Laurencia* against amoebas of the genera *Acanthamoeba* [11,12,13] and *Naegleria* [14,15]. These anti-amoeboid compounds are represented by oxasquealenoids and sesquiterpenes chemical classes, and they have been shown to induce PCD/apoptosis-like events [12,13,14,15].

As a strategy to identify marine metabolites with antiparasitic properties, we aimed to evaluate the effect of *Laurencia*-derived sesquiterpenes against *Leishmania amazonensis*. From this algae, laurequinone (**1**), laurinterol (**2**), debromolaurinterol (**3**), isolaurinterol (**4**), and aplysin (**5**) were isolated (Figure 1). The effect of the most active compound, laurequinone (**1**), was evaluated by measuring damage at the mitochondrial level, cell membrane disruption, production of reactive oxygen species (ROS), and DNA condensation. Additionally, we obtained laurequinone (**1**) by oxidation of laurinterol (**2**), the major compound of *Laurencia johnstonii*.

## 2. Results

### 2.1. Isolation, Identification, and Semisynthesis of Active Compounds

The ethanolic crude extract of *L. johnstonii* was fractionated by Sephadex LH-20 and silica gel, followed by a purification process by HPLC to yield five pure sesquiterpenes: laurequinone (**1**) [16], laurinterol (**2**) [17], debromolaurinterol (**3**) [17], isolaurinterol (**4**) [18], and aplysin (**5**) [19]. The NMR, mass spectroscopy, and optical rotation data were compared with those reported in the literature to confirm their structures. Furthermore, compound **2** was found to be the major component (80%) of the ethanolic extract. Therefore, laurinterol (**2**) was transformed by oxidation to afford the most active compound (**1**), a minor metabolite in the crude extract. The spectroscopical data of semi-synthetically obtained laurequinone (**1**) were identical to those of the natural source (Appendix A).

### 2.2. Leishmanicidal Activity

The ethanolic extract was revealed to possess antileishmanial activity with an IC_50_ value of 6.99 ± 0.6 μg/mL. After purification, the isolated sesquiterpenes **1**–**5** were evaluated against the promastigote stage of *L. amazonensis*. Additionally, they were tested against murine macrophages to analyze the toxicity of compounds and to identify those with the highest selectivity index (SI), which was calculated as the quotient between the cytotoxic concentration against murine macrophage and the inhibitory concentration against *L. amazonensis* (Table 1).

All tested compounds showed leishmanicidal activity ranging from 1.87 to 54.13 µM. However, comparing the required concentrations to inhibit 50% of the murine macrophages and the parasite, laurequinone (**1**) and aplysin (**5**) showed the best values of SI. Whilst aplysin (**5**) required higher doses to be active against promastigotes of *L. amazonensis*, laurequinone (**1**) exhibited a higher SI than the reference drug miltefosine, and a lower IC_50_ value (IC_50_ < 2 µM). 

To continue the study of the activity of the sesquiterpenes, compounds **1**–**3** were also evaluated against the intracellular stage of the parasite *L. amazonensis*. The results of IC_50_ and the SI are shown in Table 1. In this analysis, isolaurinterol (**4**) and aplysin (**5**) were discarded due to their high toxicity and low activity, respectively.

The obtained results revealed that none of the compounds tested against the intracellular amastigote form of *L. amazonensis* were more effective than the reference drug. Nonetheless, laurequinone (**1**) showed the best IC_50_ values among all the tested metabolites. Therefore, laurequinone (**1**) was selected to continue the study of the cell death mechanism induced on the parasite.

### 2.3. Cell Death Mechanisms

Studies on ATP levels, plasmatic membrane permeability, mitochondrial membrane potential, chromatin condensation, phosphatidylserine exposure, and detection of reactive oxygen species (ROS) were performed with the aim to identify the cellular effects produced by laurequinone (**1**).

Figure 2 shows that no change in ATP levels was observed for **1** when compared to untreated parasites. However, it was observed that laurequinone (**1**) induced a significant decrease in mitochondrial membrane potential after 24 h of incubation with the parasite. The mitochondrial membrane potential assay was performed with JC-1 reagent, in which the component changes from a monomer to a dimer depending on the mitochondrial potential, each one of a different color (monomers are excited at lower wavelengths and emit green, while the aggregates are excited at a higher wavelength and emit red), thus allowing the changes to be detected. This phenomenon of a reduction of the mitochondrial membrane potential is associated with cell death by apoptosis, also known as programmed cell death, which is a controlled and orderly mechanism that does not induce an inflammatory response. Sodium azide, a well-known mitochondrial inhibitor, was added as a positive control.

To determine the presence of reactive oxygen species (ROS) in *L. amazonensis* promastigotes when exposed to laurequinone (**1**), we employed a probe called CellROX Deep Red. This method allowed us to visualize and analyze the results using fluorescence imaging (Figure 3).

We used untreated parasites as negative controls, which were expected to display minimal or no red fluorescence when observed under a microscope. In contrast, when *L. amazonensis* parasites were treated with laurequinone (**1**) at its IC_90_ concentration for 24 h, we observed a noticeable increase in visual fluorescence of CellROX Deep Red. This indicated that the compound induced the accumulation of ROS inside the cytoplasm of the promastigotes, as evidenced by the enhanced red fluorescence signal. This may be because the parasites are unable to eliminate the ROS produced, and therefore, they accumulate and generate oxidative stress in the cells.

Studying the effect of the compound on the cytoplasmic membrane of the parasite, we observed a characteristic phenomenon that is often associated with programmed cell death, namely the exposure of phosphatidylserine residues on the outer surface of the membrane. Figure 4 shows a visual representation of this phenomenon, where cells stained green indicate the presence of exposed phosphatidylserine residues.

When the parasites were incubated with laurequinone (**1**), we observed that approximately 30% of the cell population exhibited this characteristic phenomenon. In contrast, untreated control cells did not show such an event, as the percentage of cells with exposed phosphatidylserine residues was zero. To validate our findings, we included miltefosine as a positive control, which is known to induce similar apoptotic effects.

Another phenomenon studied was the condensation of nuclear chromatin, which is also indicative of this type of controlled death. Hoechst, a blue dye that stains condensed nuclear chromatin, was used for this purpose. The commercial kit also uses propidium iodide, which stains the cytoplasm of dead cells red. The results were negative, so no chromatin condensation was observed inside the parasites treated with laurequinone (**1**).

To further investigate the integrity of the plasma membrane to ensure that it had not been compromised or altered, an additional assay was performed. This assay was intended to rule out necrotic death, which is characterized by membrane rupture. As a positive control, we treated the cells with 0.5% Triton X-100 for 3 h, as it is known to induce membrane permeabilization. To assess the state of the plasma membrane, we used a dye called SYTOX Green. This dye is normally impermeable to the cytoplasmic membrane of cells. However, if the membrane becomes permeable or compromised, the dye can enter the cell and bind to DNA, resulting in a strong green fluorescence. Figure 5 shows the results of this analysis. Since no significant green fluorescence was observed, this indicated that the cells did not show increased permeability and ruled out necrosis as a mode of cell death. This finding further supported the conclusion that the observed effects were not due to membrane damage but were attributed to other mechanisms.

## 3. Discussion

Natural products from marine algae are known for their broad spectrum of bioactivities. It has been suggested that over 150 species out of the more than 30,000 known algal species have been evaluated against *Leishmania* spp., most of them corresponding to red seaweed. Interestingly, species from *Laurencia* complex are considered among the most relevant macroalgae for antileishmanial drug discovery. However, only a few studies have been continued with the chemical analysis to identify the active compounds, most of them represented by terpenes [20].

With the aim to contribute to the field of antiparasitic agents, in this work, laurequinone (**1**), laurinterol (**2**), debromolaurinterol (**3**), isolaurinterol (**4**), and aplysin (**5**) were isolated from red algae *L. johnstonii.* Laurinterol (**2**) has been a widely studied compound. It has exhibited insecticidal, repellent [21], antifouling [22], antibacterial [23], antimycobacterial [24], cytotoxic [25,26], antitumoral [27], and anti-amoebic properties [11,14]. It has been suggested to trigger PCD-like events by inhibition of the Na^+^/K^+^-ATPase sodium–potassium ion pump [14,15,27]. However, whereas the anti-amoebic activity of sesquiterpenes **2**–**5** has been previously reported by our research group [11,14,15], the biological properties of compound **1** have hardly been studied. Despite that these metabolites were first isolated decades ago, to the best of our knowledge, they have not been evaluated against *Leishmania* spp.

Herein, the anti-kinetoplastid activity of compounds **1**–**5** against *L. amazonensis* has been evaluated, with IC_50_ values ranging between 1.87 and 54.13 µM. The analysis of IC_50_ and CC_50_ values confirmed laurequinone (**1**, IC_50_ 0.43 µg/mL, SI 22.5) as the most promising molecule. It is the only tested compound to satisfy the activity criteria to be considered as a leishmanicidal hit (IC_50_ < 1 µg/mL and SI > 20) [28,29], which also improves the performance of miltefosine (SI 11.1) against promastigotes of *L. amazonensis.* Young and Pesce [10] reviewed the antileishmanial potential of over 45 marine metabolites isolated from different natural sources, considering as biologically active those compounds with activity below 25 µM against both promastigotes and amastigotes. According to this criterion, debromolaurinterol (**3**) and isolaurinterol (**4**), with an IC_50_ < 15 µM, and laurequinone (**1**), with an IC_50_ < 2 µM, can be considered as antileishmanial lead compounds.

However, two crucial factors pose limitations to the advancement of studies on potential lead compounds. Firstly, the active compounds must exhibit a selectivity level that is at least comparable to that of the reference drug. Secondly, the evaluation of activity should be performed on intracellular amastigotes, which are the relevant stage for leishmaniasis pathogenesis [20,30]. Hence, we assessed the effect of compound **1** on *L. amazonensis* amastigotes. However, the reference drug demonstrated superior selectivity for the intracellular stage of the parasite. Álvarez-Bardón et al. [30] conducted a review on marine-derived antiparasitic compounds, with a specific focus on those exhibiting activity against the intracellular amastigote form of *Leishmania* spp. Among the metabolites isolated from algae, the most potent compound was a diterpene derived from a brown Dictyotaceae seaweed, which displayed an IC_50_ value of 11.0 µM against *L. amazonensis* amastigotes [31].

Plumbagin, a natural naphthoquinone derived from *Pera benensis*, was tested against *L. donovani*, demonstrating an IC_50_ value of 0.34 μM for promastigotes and 0.21 μM for axenic amastigotes. It has been revealed that the compound’s mechanism of action involves non-competitive inhibition of trypanothione reductase, a vital enzyme in *Leishmania*’s redox homeostasis. This inhibition leads to an elevation in reactive oxygen species and an alteration of the redox balance [32]. Similarly, treatment with laurequinone (**1**) triggers an increase in reactive oxygen species, as depicted in Figure 3. Hence, it is plausible that laurequinone (**1**) operates via a comparable mechanism, potentially inhibiting trypanothione reductase or another key enzyme responsible for redox homeostasis.

Although numerous marine natural products have shown antileishmanial activity, none of them are currently available on the market. In this context, we successfully obtained laurequinone (**1**) by oxidizing laurinterol (**2**), following previous studies that achieved laurequinone through the transformation of debromolaurinterol [16] and cupalaurenol [33]. Considering the potential of laurequinone (**1**) as a lead compound for the development of antileishmanial agents, it was crucial to investigate the cell death mechanisms induced by this compound in *L. amazonensis* promastigotes. However, we did not observe chromatin condensation, and several PCD-related events were examined, including the accumulation of reactive oxygen species (ROS), a decrease in mitochondrial membrane potential without affecting ATP levels, and the exposure of phosphatidylserine on the cell surface without alterations in the plasma membrane permeability.

Considering the molecular structure of the tested compounds, the best antileishmanial activity was exhibited by the sesquiterpene quinone. A plausible mechanism for DNA damage involves electron transfer chemistry, such as that proposed for avarone. Thus, a one-electron reduction of **1**, probably catalyzed by enzymes such as trypanothione reductase, yields laurequinone semiquinone radicals, which can transfer an electron to molecular oxygen (O_2_) to produce superoxide (O_2_^●−^) (Figure 6). Both the superoxide and laurequinone semiquinone radical anions can generate hydroxyl radicals, which is known to cause DNA strand breaks [34].

This proposal may be supported by the study of Imperatore et al., who investigated the antiparasitic potential of the marine sesquiterpenoids, avarol and avarone, and the derivative thiazoavarone. According to computational studies’ calculations, they suggested that the antiparasitic effect could be produced by a toxic semiquinone radical species, starting both from quinone- and hydroquinone-based compounds. Interestingly, these compounds exhibited a higher SI for the amastigote form compared to the promastigote form; however, the reference drug amphotericin B was still better according to the IC_50_ and SI [35]. On the other hand, due to its quinone structure, the possibility that it could act as a Michael acceptor cannot be discarded.

## 4. Materials and Methods

### 4.1. Extraction and Isolation of Metabolites

The ethanolic extract of *L. johnstonii* collected in Baja California Sur, Mexico, was fractionated by Sephadex LH-20 (500 × ∅ 70 mm, CH_3_OH, 100%), and 5 fractions were obtained. Fraction 4 was chromatographed over a silica gel column (250 × ∅ 50 mm) using a step-gradient from *n*-hexane (*n*-Hex) to ethyl acetate (EtOAc) to yield 14 fractions. The fraction enriched in sesquiterpenes was separated by HPLC on a normal-phase column (Luna 5 μm Silica 100 Å, 250 × ∅ 10 mm), using a step-gradient of *n*-hexane/EtOAc to yield the pure compounds laurequinone (**1**) [16], laurinterol (**2**) [17], debromolaurinterol (**3**) [17], isolaurinterol (**4**) [18], and aplysin (**5**) [19].

### 4.2. Conversion of Laurinterol (2) into Laurequinone (1)

Laurequinone (**1**) was obtained by oxidation of laurinterol (**2**) according to the methodology described by Ichiba and Higa [33], with some modifications. Briefly, **2** (45 mg, 0.15 mmol) was dissolved in 80% acetic acid (2 mL), and 500 μL of 25% chromium trioxide solution in acetic acid was added. The reaction was left under magnetic stirring for 2 h at 0 °C and extracted with dichloromethane after the addition of water. The reaction mixture was separated on a normal-phase open column (Silicagel, 150 × ∅ 20 mm) eluted with *n*-Hex/EtOAc (99:1), and finally purified by HPLC on a normal-phase column (Luna 5 μm Silica 100 Å, 250 × ∅ 10 mm) eluted with *n*-Hex/EtOAc (99:1) to yield **1** (15.8 mg, 0.07 mmol).

### 4.3. Cell Cultures

*Leishmania amazonensis* (MHOM/BR/77/LTB0016) promastigotes were cultured in RPMI 1640 medium (Gibco, Waltham, MA, USA), supplemented with 10 % FBS at 26 °C. Additionally, for maintenance, the promastigotes were cultured in SND medium (Sigma-Aldrich, Darmstadt, Germany) supplemented with 10 % FBS at 26 °C. Both media were supplemented with gentamicin at 10 µg/mL.

Murine macrophages (J774A.1) ATCC TIB-67 were cultured in RPMI, supplemented with 10% FBS at 37 °C and 5 % CO_2_ atmosphere.

### 4.4. Antileishmanial Activity

Compounds **1**–**5** were tested against promastigotes of *Leishmania amazonensis* using the colorimetric assay based on the Alamar Blue^®^ reagent. In a 96-well plate, serial dilutions of the compounds and the promastigote form of the parasite (10^6^ cells/mL) were added at a final volume of 200 µL. After 48 h of incubation, 10% Alamar Blue^®^ reagent was added and fluorescence was measured 24 h later, using the EnSpire Multimode Plate Reader^®^ at 544 nm excitation and 590 nm emission to calculate the 50% inhibitory concentration (IC_50_) [36].

The most active compound against the promastigote stage was tested on the intracellular amastigote form. Briefly, in a 96-well plate, murine macrophages (2 × 10^5^ cells/mL) were incubated in RPMI medium at 37 °C and 5 % CO_2_ atmosphere for 2–4 h. Promastigote forms of *L. amazonensis* (2 × 10^6^ cells/mL) were added to reach a 1:10 macrophage:parasite ration, and after 24 h of incubation, the cells were washed to discard the not-introduced parasites. Serial dilutions of the compound were added in 100 µL of RPMI medium. Following 24 h of incubation, the plate was washed, and the cells were broken with SDS 0.05% for 30 s. Fresh medium was added, and 48 h later, 10% Alamar Blue^®^ reagent was added. The plate was incubated for an additional 24 h at 26 °C and the fluorescence was read by using the EnSpire Multimode Plate Reader^®^ at 544 nm excitation and 590 nm emission to calculate the IC_50_ [37].

### 4.5. Cytotoxic Activity

In a 96-well plate, macrophages (2 × 10^5^ cells/mL) were added. After completing the adherence (2–4 h), macrophages were incubated with serial dilutions of compounds **1**–**5**. Then, 10% Alamar Blue^®^ was added, and fluorescence was read following 24 h of incubation at 37 °C and 5% CO_2_ using the EnSpire Multimode Plate Reader^®^ at 544 nm excitation and 590 nm emission to calculate the 50% cytotoxic concentration (CC_50_). The selectivity index (SI) was calculated as the ratio of CC_50_ and IC_50_ [38].

### 4.6. Cell Death Mechanisms

All events related to the mechanisms of cell death analysis of the most active compound were performed in a 96-well plate by incubation of promastigotes (10^6^ cells/mL), with the 90% inhibitory concentration (IC_90_) in RPMI medium to reach a 200 µL final volume. After incubation for 24 h at 26 °C, samples were centrifuged (3000 rpm, 10 min, 4 °C) and resuspended in 50 µL of PBS buffer. Different events involved in parasite cell death were studied following the manufacturer’s instructions for each kit.

#### 4.6.1. ATP Level Analysis

ATP levels were measured using the CellTiter-Glo^®^ Luminescent Cell Viability Assay kit (Promega, Fitchburg, WI, USA) and the EnSpire Multimode Plate Reader^®^ (PerkinElmer, Waltham, MA, USA). After 24 h of incubation with laurequinone (**1**), the pellet was resuspended in the buffer and the kit. After 2 min of agitation and 10 min of incubation at room temperature in the dark, the luminescence was measured with the EnSpire Multimode Plate Reader^®^. The results were plotted against the negative control of untreated cells. Sodium azide (20 mM) was added as a positive control [39].

#### 4.6.2. Mitochondrial Membrane Potential Analysis

The JC-1 Mitochondrial Membrane Potential Assay Kit^®^ (Cayman Chemical, Ann Arbor, MI, USA) was used to determine whether an alteration in the mitochondrial potential occurred during cell death. The parasites treated with laurequinone were incubated with the staining solution for 20 min at 26 °C in the dark. Green and red fluorescence were measured using the EnSpire Multimode Plate Reader^®^ (PerkinElmer). The results were expressed as the percentage relative to the untreated cells of the red (J-aggregates 540/570 nm)/green ratio (J-monomers 485/535 nm). The membrane-permeant JC-1 dye is widely used in apoptosis studies to monitor mitochondrial health. JC-1 dye exhibits potential-dependent accumulation in mitochondria, indicated by a green fluorescence emission for the monomeric form of the probe, which shifts to red with a concentration-dependent formation of red fluorescent J-aggregates. Consequently, mitochondrial depolarization is indicated by a decrease in the red/green fluorescence intensity ratio. Sodium azide (20 mM) was added as a positive control [40].

#### 4.6.3. Phosphatidylserine Externalization

A double-staining assay with annexin V/propidium iodide (PI) was performed using the Tali^®^ Apoptosis Kit–Annexin V Alexa Fluor^®^ 488 (Invitrogen^TM^, Waltham, MA, USA). Briefly, after incubating the treated parasites for 24 h, they were washed with the buffer and incubated with annexin V for 20 min. Promastigotes were centrifuged and resuspended in buffer containing PI and incubated for 3 min. Finally, the stained cells were loaded into a slide and inserted in a TALI^TM^ image-based cytometer (Life Technologies, Carlsbad, CA, USA). Data were expressed in percentages, divided into three groups: apoptotic cells (green fluorescence), dead cells (red or red and green fluorescence), and live cells (no fluorescence), provided by TALI^®^ data acquisition and analysis software (Life Technologies) [40].

#### 4.6.4. Reactive Oxygen Species Analysis

Oxidative stress is the result of an imbalance between the production of reactive oxygen species (ROS) and the cells’ ability to remove them. The CellROX^®^ Deep Red Reagent (ThermoFisher Scientific, Waltham, MA, USA) was used to measure the oxidative stress. CellROX™ Deep Red reagent does not fluoresce when in the reduced state and fluoresces upon oxidation by reactive oxygen species. Treated parasites were incubated and resuspended in the buffer, the kit was added, and the plate was incubated for 30 min at 26 °C in the dark. Oxidative stress is represented as red fluorescence in the images obtained with the Cy5 light cube (excitation 644 nm/emission 665 nm) of the EVOS^®^ FL Cell Imaging System (ThermoFisher Scientific, Waltham, MA, USA). Untreated parasites were used as a negative control, and hydrogen peroxide (600 mM) was used as a positive control [36].

#### 4.6.5. Chromatin Condensation Analysis

The Invitrogen Chromatin Condensation/Dead Cell Apoptosis Kit, Hoechst 33342/propidium iodide (PI), was used to determine whether chromatin condensation was produced in the treated parasites. After adding the kit, the parasites were incubated for 30 min at 26 °C in the dark. The results were observed under the EVOS fluorescence microscope using DAPI (excitation 350 nm/emission 461 nm for Hoechst) and RFP (excitation 535 nm/emission 617 nm for PI) light cubes. The obtained images were divided into three types of results: low fluorescence of both stains when the cells were alive, intense fluorescence when the cell had condensed chromatin (programmed cell death), and intense red fluorescence when the cell had died [14].

#### 4.6.6. Plasma Membrane Permeability

The SYTOX Green assay (Life Technologies, Madrid, Spain) was used to determine the damage in the plasma membrane permeability of treated cells. Parasites were incubated for 24 h with the compound and the stain was added and incubated for 15 min. Cells were observed in the EVOS™ M5000 Imaging System (Invitrogen by Thermo Fisher Scientific) at an excitation wavelength of 504 nm and an emission wavelength of 523 nm [39]. Triton X-100, a well-known non-ionic surfactant, was added at 0.5% for 3 h before staining, as a positive control.

### 4.7. Statistical Analysis

All experiments were performed in duplicate in three independent experiments and data were presented as the mean ± standard error (SE). By using SigmaPlot 12.0 software, statistical differences between means were tested using a one-way analysis of variance (ANOVA) using Tukey’s test. A significance level of *p* < 0.05 was used.

## 5. Conclusions

Overall, we successfully identified five sesquiterpenes isolated from the alga *Laurencia johnstonii* with potent leishmanicidal activity, that allowed us to identify new candidate molecules for the treatment of leishmaniasis. The isolated compounds induced several physiological and morphological changes in *L. amazonensis* promastigotes. Changes in mitochondrial membrane potential, ROS production, phosphatidylserine externalization, and cell shrinkage, coupled with the absence of changes in membrane permeability, were observed in the parasites that were treated with the aforementioned molecules. The most relevant finding of the present study was that laurequinone (**1**) induced programmed cell death in *L. amazonensis*, sharing several phenotypic features with other cases of programmed cell death in metazoans. Laurequinone (**1**) was revealed as a promising candidate against leishmaniasis. These results support future research to conduct further studies to unveil the target of these sesquiterpenes and lead to the optimization of a future treatment for leishmaniasis.

## 6. Patents

Protected under registration numbers: P202230582 (Spanish Patent and Trademark Office, OEPM) and MX/a/2022/011373 (Mexican Institute of Intellectual Property, IMPI).

## Figures and Tables

**Figure 1 marinedrugs-21-00333-f001:**
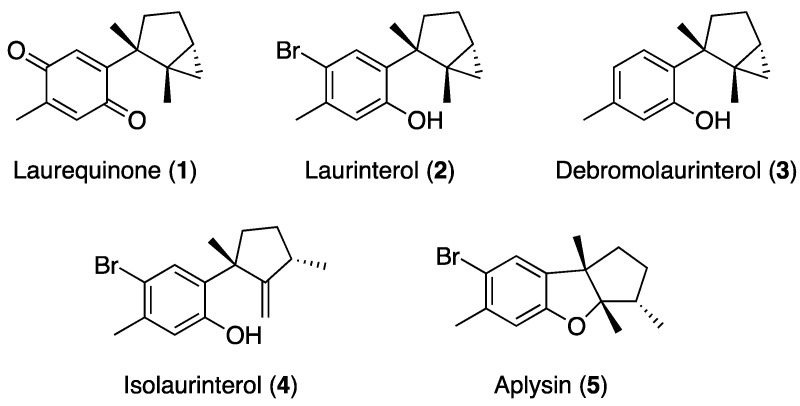
Chemical structure of the pure compounds isolated from *Laurencia johnstonii*.

**Figure 2 marinedrugs-21-00333-f002:**
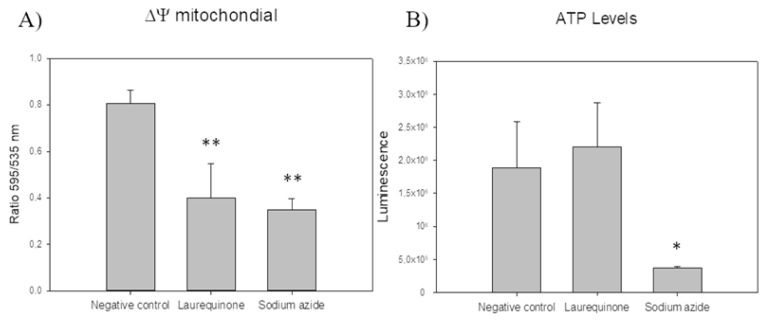
(**A**) Changes in the mitochondrial membrane potential (ΔΨm) and (**B**) ATP levels of *Leishmania amazonensis* promastigotes after 24 h of incubation with the IC_90_ of laurequinone (**1**). Error bars represent the standard deviations (SD). Each data point indicates the mean of the results of three measurements, (**) *p* < 0.01, (*) *p* < 0.05.

**Figure 3 marinedrugs-21-00333-f003:**
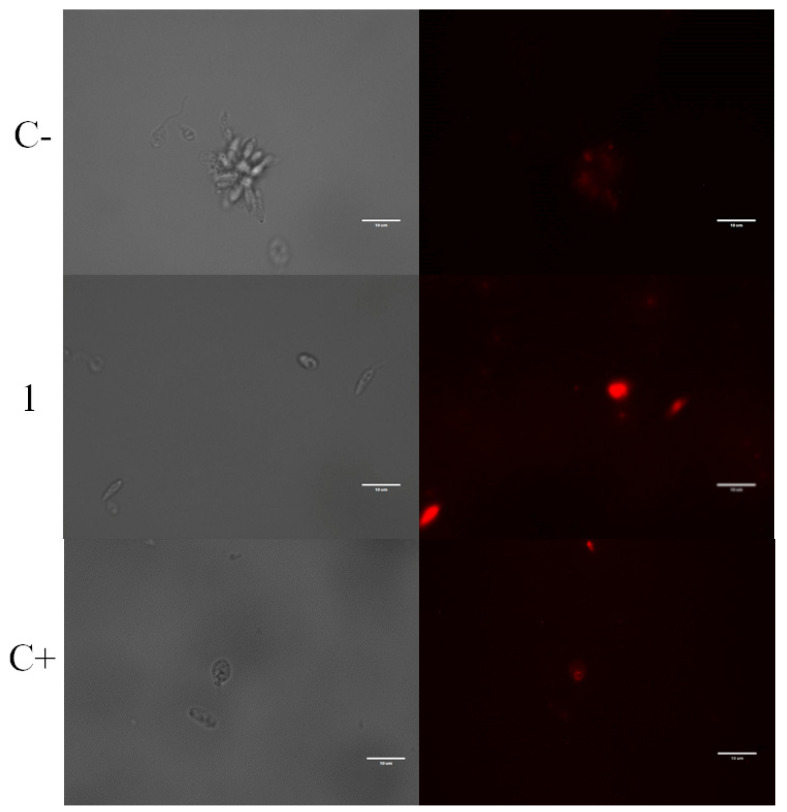
CellROX Deep Red staining. Results after 24 h of incubation of *L. amazonensis* promastigotes with the IC_90_ of laurequinone (**1**). C−: Negative control. C+: Parasites treated with hydrogen peroxide. Images were captured using an EVOS FL Cell Imaging system (Thermo Fisher Scientific) (100×). Scale bars: 10 µm.

**Figure 4 marinedrugs-21-00333-f004:**
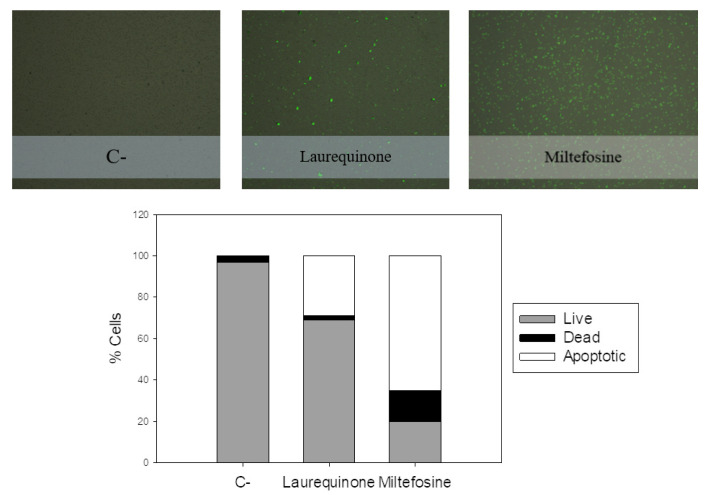
Results of the phosphatidylserine exposure after 24 h of incubation of *L. amazonensis* promastigotes with the IC_90_ of laurequinone (**1**). Images were captured using a TALI image-based cytometer (Invitrogen).

**Figure 5 marinedrugs-21-00333-f005:**
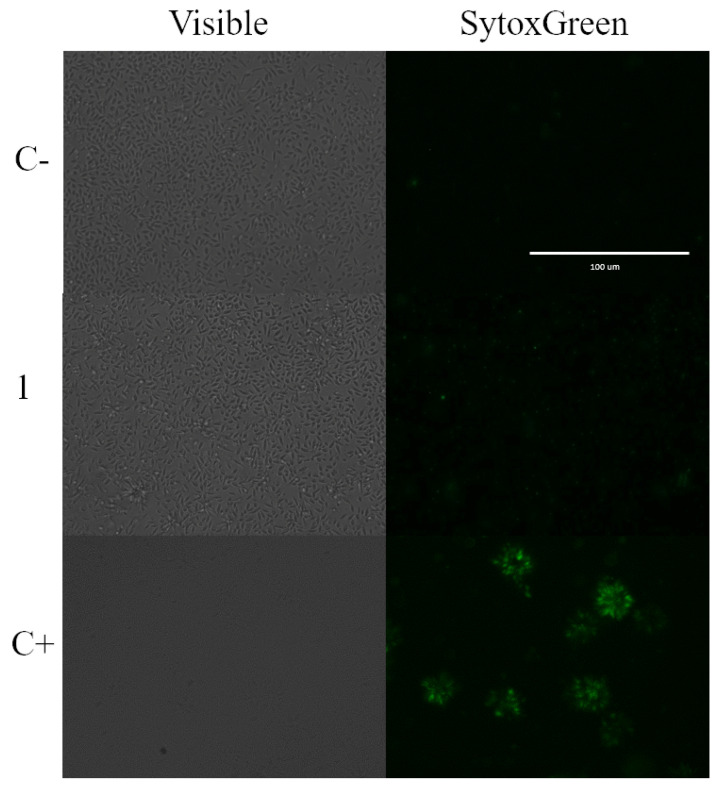
SYTOX Green staining. Results after 24 h of incubation of *L. amazonensis* promastigotes with the IC_90_ of laurequinone (**1**). C−: Negative control. C+: Treated with Triton X-100. Images were captured using an EVOS FL Cell Imaging system (Thermo Fisher Scientific) (40×). Scale bar: 100 µm.

**Figure 6 marinedrugs-21-00333-f006:**
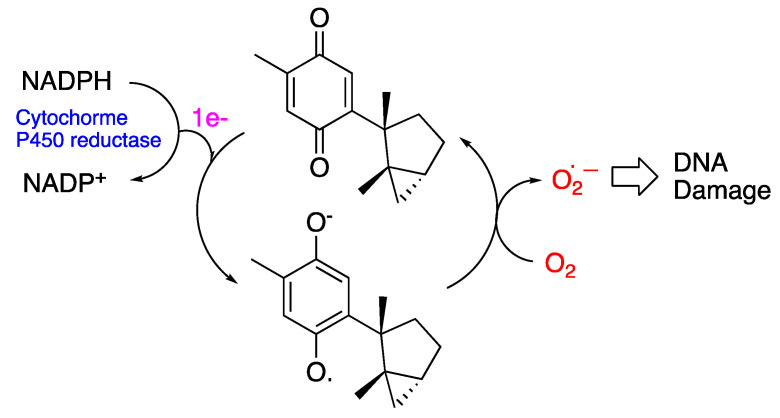
Plausible electron transfer mechanism for DNA damages by laurequinone (**1**).

**Table 1 marinedrugs-21-00333-t001:** Effect of sesquiterpenes **1**–**5** against murine macrophages (CC_50_, µM) and *L. amazonesis* promastigotes and amastigotes. SI: selectivity index. Nd: not determined.

Compound	CC_50_, µM	IC_50_, µM Promastigotes	SI	IC_50_, µM Amastigotes	SI
Laurequinone (**1**)	42.07 ± 1.30	1.87 ± 0.17	22.5	16.72 ± 6.08	2.5
Laurinterol (**2**)	80.11 ± 7.79	34.45 ± 6.44	2.3	34.72 ± 4.06	2.3
Debromolaurinterol (**3**)	70.13 ± 12.94	12.48 ± 0.05	5.6	24.32 ± 5.09	2.9
Isolaurinterol (**4**)	24.56 ± 2.37	10.09 ± 1.69	2.4	Nd	Nd
Aplysin (**5**)	1096.44 ± 40.65	54.13 ± 6,44	20.3	Nd	Nd
Miltefosine	72.18 ± 3.07	6.48 ± 0.25	11.1	3.12 ± 0.29	23.2

## Data Availability

Not applicable.

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
