# Peer review of "Laurequinone, a Lead Compound against Leishmania"

_marinedrugs, 2023, doi:10.3390/md21060333_

Round 1

Reviewer 1 Report

In this study, five different sesquiterpenes were isolate and identified from red alga Laurencia johnstonii.These compounds

were tested in vitro against the promastigote and amastigote forms of Leishmania amazonensis. laurequinone(1) showed certain potential against Leishmania.However, it is of little practical significance to detect the activity of several compounds on promastigote and the mechanism of drug action, why not directly detect the action mechanism of drugs on amastigote? Because the promastigote is parasitic on the vector insect sandfly, Only the amastigote phase is parasitic in human macrophages. The mechanism of action of compounds on  promastigote  is not indicative of the mechanism of action on amastigote.In addition, the description of some methods and results in the article is not detailed enough to support the conclusions proposed by the author.

1.  Line47-48 The introduction of leishmaniasis is not clear, its transmission vector is insect sandfly, there are no multiple vectors.

2. Important references are not marked, such as Line45-46. Where are the data sources of the number of people infected with leishmaniasis? Is it the latest data? Line58 What references does the hypothesis about the mechanism of action of miltefosine come from?

3. In the part of methods, does the word parasite refer to the   promastigote or amastigote forms of Leishmania amazonensis.?It should be clearly stated.

4. Line 151:What does the sentence in parentheses mean?Figure 1 shows a bar chart without color representation.

5. Figure 3 The number of cells in the figure is too small, and the author's description of the results is not detailed enough to support the author's conclusion.

6. Figure 5 Why is there a negative result after PI staining? After the action of the drug, no matter which way the cells die, PI staining will be positive,  the negative result means that these cells did not die, and the drug did not exert against Leishmania activity.

7. The statistical analysis describes only three repetitions of the experiment, and how many samples are repeated each time?

Reviewer 2 Report

The manuscript entitled "Laurequinone, a lead compound against Leishmania" Title, abstract and overall rationale of work is written satisfactory. There are major concerns, which needs to be addressed before publication.

1) In the abstract: This part should be write concise way and author wrote material method part more instead of the result part. Author need to revise. Secondly, author need to write all five compound IC50 value in the abstract part.

2) In the introduction section: Author also need to mention death rate due to leishmaniasis happen in every year.

3) Why author written compound name laurequinone (1), laurinterol (2) and other with number. Compound name is different so I suggest no need to write compound name and number together.

4) Results section: Line no. 107 author must be write italic the species name and also check whole manuscript and correct it.

The quality of figure 2 is not good and author need to increase resolution.

5) This results is very primary and only compound 1 is showing good inhibitory activity only promastigote stage however amastigote stage this compound is not effective. Author need to explain more details about this? I also recommend author need to do in-vivo study to confirm this results.

6) How many times this experiment repeat and after repetition the IC50 value is nearby or drastically changes. Author need to add all data.

7) Discussion section is written well and details however I suggest the author they need to write concise way.

8) Material method section is written good and all information is available.

9)  Some references are too long and author need to revise for example reference no14, 15, 16 and other. I suggest author to revise if other latest manuscript is available in the same information.

Reviewer 3 Report

The article is novel, and well-designed; The authors can find some comments in the following.

Line 312,  (106 cells/mL) should be edited.

In amastigote assay they used Alomar blue and fluorometric assay, how did they can discriminate the viability of intracellular amastigote  from J774 cell line. The Inhibitory concentration is for both MQ and parasite coculture.  

The quality of Figures 3, 4, and 5 are poor.

Round 2

Reviewer 2 Report

The authors have addressed all the concerns raised in the previous version of the manuscript and the quality has much improved after incorporating required modifications. Therefore, the manuscript may be considered for publication in this Journal.